# Characterization of Diarreaghenic *Escherichia coli* Strains Isolated from Healthy Donors, including a Triple Hybrid Strain

**DOI:** 10.3390/antibiotics11070833

**Published:** 2022-06-21

**Authors:** Evelyn Méndez-Moreno, Liliana Caporal-Hernandez, Pablo A. Mendez-Pfeiffer, Yessica Enciso-Martinez, Rafael De la Rosa López, Dora Valencia, Margarita M. P. Arenas-Hernández, Manuel G. Ballesteros-Monrreal, Edwin Barrios-Villa

**Affiliations:** 1Departamento de Ciencias Químico Biológicas y Agropecuarias, Universidad de Sonora, Unidad Regional Norte, Campus Caborca. Av. Universidad e Irigoyen S/N, col. Eleazar Ortiz, H. Caborca CP 83621, Sonora, Mexico; a217200984@unison.mx (E.M.-M.); liliana.caporal@unison.mx (L.C.-H.); pablo.mendez@unison.mx (P.A.M.-P.); yessica.enciso@unison.mx (Y.E.-M.); rafael.delarosa@unison.mx (R.D.l.R.L.); dora.valencia@unison.mx (D.V.); 2Posgrado en Microbiología, Centro de Investigación en Ciencias Microbiológicas, Instituto de Ciencias, Benemérita Universidad Autónoma de Puebla, Ciudad Universitaria, Puebla CP 72570, Pue, Mexico; margarita.arenas@correo.buap.mx

**Keywords:** *Escherichia coli*, pathotypes, healthy donors, antimicrobial resistance

## Abstract

*Escherichia coli* is a well-recognized inhabitant of the animal and human gut. Its presence represents an essential component of the microbiome. There are six pathogenic variants of *E. coli* associated with diarrheal processes, known as pathotypes. These harbor genetic determinants that allow them to be classified as such. In this work, we report the presence of diarrheagenic pathotypes of *E. coli* strains isolated from healthy donors. Ninety *E. coli* strains were analyzed, of which forty-six (51%) harbored virulence markers specifics for diarrheagenic pathotypes, including four hybrids (one of them with genetic determinants of three DEC pathotypes). We also identified phylogenetic groups with a higher prevalence of B2 (45.6%) and A (17.8%). In addition, resistance to sulfonamides (100%), and aminoglycosides (100%) was found in 100% of the strains, with a lower prevalence of resistance to cefotaxime (13.3%), ceftriaxone (12.2%), fosfomycin (10%), and meropenem (0%). All analyzed strains were classified as multidrug resistant. Virulence genes were also investigated, which led us to propose three new virotypes. Among the virulence traits observed, the ability to form biofilms stands out, which was superior to that of the *E. coli* and *Staphylococcus aureus* strains used as positive controls.

## 1. Introduction

*E. coli* is a gastrointestinal tract inhabitant of humans, forming part of the gut microbiota. They are commonly beneficent in protecting the gastrointestinal epithelium from other microorganisms; they provide B12 and K vitamins and frequently activate the immune system [1]. Nevertheless, eventually, those commensal strains might acquire some virulence traits, which can confer them with better adaptation to several niches, leading to them establishing and causing a large number of diseases [2].

Disease-associated *E. coli* strains have been classified into pathotypes, also divided into diarrheagenic *E. coli* (DEC, associated with diarrhea) and extraintestinal pathogenic *E. coli* (ExPEC) [3]. DEC strains are subdivided into six well-recognized pathotypes (enteroinvasive *E. coli*, EIEC; enteropathogenic *E. coli*, EPEC; enterotoxigenic *E. coli*, ETEC; diffusely adherent *E. coli*, DAEC, enterohemorrhagic *E. coli*, EHEC; and enteroaggregative *E. coli*, EAEC), but some hybrid variants have also been reported, suggesting hypervirulence [1]. Those DEC pathotypes exhibit genetic determinants associated with their participation in disease development and can be used as molecular markers of the pathotype. An important phenotypic trait associated with each pathotype is its adherence pattern, which is directly linked to fimbrial or non-fimbrial adhesins [2,4,5,6].

Although the effect of antibiotic therapy on infections caused by pathogenic *E. coli* is recognized as uncertain, clinical practice guidelines in Mexico include the use of sulfamethoxazole/trimethoprim, quinolones, and cephalosporins [7]. However, increasing reports have emerged reporting antimicrobial resistance in *Enterobacteriaceae* members, including *E. coli*, which exhibit several mechanisms to avoid antibiotic effects, frequently encoding into mobilizable elements such as plasmids, transposons, or integrons [8,9,10,11,12,13,14,15].

DEC strains are not exclusively human pathogens; they can be present in several reservoirs, including wildlife or domestic animals [16,17,18]. In this sense, Clermont et al. proposed the use of a classification scheme based on the detection of a group of genes to establish phylogenetic groups [19]. Based on this scheme, phylogroup A has been linked with commensal strains, D2 has been associated with environmental strains, and B1, B2, E, and D1 have been related to pathogenic strains [20].

The role of DEC strains is widely recognized, and individuals with diarrheal illness can be easily monitored [21]. Water and food can be treated to reduce or eliminate the microorganism, but the role of asymptomatic carriers remains unclear. In this paper, we report the presence of *E. coli* strains that exhibit virulence determinants and genetic markers associated with diarrheagenic pathotypes isolated from healthy humans, as well as their resistance patterns and the genes they harbor.

## 2. Results

### 2.1. Strains Identification

From 50 donors’ stool samples collected and inoculated on MacConkey agar plates, 189 lactose-positive and 13 lactose-negative colonies were selected. By biochemical characterization, we identified 100 strains as *E. coli*, 12 with a 90% of probability of being *E. coli* or *L. adecarboxylata*, and 11 as *L. adecarboxylata*; the remaining 79 corresponded to *Buttiauxella* sp., *Citrobacter* spp., *Edwardsiella* sp., *Enterobacter* spp., *Klebsiella* sp., *Kluyvera* spp., *Leminorella* sp., *Morganella* sp., *Pantoea* spp., *Proteus* sp., *Providencia* spp., *Raoultella* spp., *Salmonella* spp., *Serratia* spp., *Shigella* spp., and *Yersinia* spp.

The strains biochemically identified as *E. coli* or *L. adecarboxylata* were tested for molecular identification using the *ybbW* gene, which is conserved in *E. coli*, but absent in *L. adecarboxylata*, confirming the identity of 90 *E. coli* strains. All identified strains corresponded to those frequently reported as forming part of the gastrointestinal tract.

### 2.2. Phylogenetic Group Determination

We determined phylogenetic groups of 90 analyzed strains according to the scheme of Clermont et al. [19]. The more prevalent phylogenetic group was B2 (45.5%, *n* = 41), followed by A (17.8%, *n* = 16), clade I or II (16%, *n* = 18), and the distribution of the remaining three phylogroups were F (5%), C (3%) and B1 (1%). Interestingly, except for phylogroup A, which is associated with commensal strains, phylogroups frequently related to *E. coli* strains causing human infections were identified; however, an important number of strains (*n* = 36) were not associated with a specific phylogenetic group, which might be explained by the non-pathogenic origin of the isolates.

### 2.3. E. coli Isolates from Healthy Humans, Harbor Several Resistant Traits

The 90 *E. coli* strains were tested using agar dilution methodology for resistance phenotypes. A high percentage of resistance to gentamicin, sulfonamides, and nitrofurans was observed (100%, 100%, and 98%, respectively), followed by ampicillin resistance (94.4%), nalidixic acid (86.7%), ciprofloxacin (81.1%), amikacin (54.4%), tetracycline (48.9%), cefoxitin (15.56%), cefotaxime (13.3%), chloramphenicol (12.2%), ceftriaxone (12.2%), fosfomycin (10%), and meropenem (0%). All analyzed isolates were resistant to five or more antibiotic categories, classifying them as multidrug-resistant strains. Based on these results, the presence of extended-spectrum β-lactamases (ESBL) associated genes was investigated in the strains showing resistance to cefotaxime. Twelve strains were evaluated; the more prevalent ESBL gene was *bla*_CTX-M1_, which was identified in seven strains (58.3%) (*n* = 12 analyzed isolates). Interestingly, two strains showed co-occurrence of the *bla*_CTX-M2_ gene that has been reported to be associated with *bla*_CTX-M15_; when we looked for the CTX-M15 variant of the gene, it was absent (Table 1). In addition, we analyzed each strain harboring CTX genes for their Minimum Inhibitory Concentration (MIC). Those isolated from Donor 9 showed the highest levels of resistance for the three cephalosporins tested (256 µg/mL), even they were carrying one or two associated genes, while the strains isolated from Donor 12, showed a more heterogeneous behavior, with MIC of 128 µg/mL for FOX and CTX in two strains and for CRO in one strain, and 64 µg/mL for FOX and CTX in another one.

### 2.4. High Prevalence of aEPEC Pathotype between E. coli Strains Isolated from Healthy Donors

Diarrheagenic pathotypes were investigated between the confirmed *E. coli* strains; 45.5% (*n* = 41) of the strains were positive for the atypical EPEC (aEPEC) pathotype. Interestingly, three hybrid pathotypes were found, two strains harbor *bfpA* and LT genes (aEPEC/ETEC pathotype), and one isolate carries *bfpA*, and *daaE* genes (aEPEC/DAEC pathotype); a surprising finding was the sample EC-36.3, which showed genetic determinants for aEPEC/DAEC/STEC simultaneously. The remaining pathotypes were not identified in the isolated strains (Table 1).

### 2.5. Adherence Patterns of E. coli Diarrheagenic Pathotypes Identified

To establish a relationship between the pathotype identified and the adherence traits of the strains, those positive to a specific pathotype were tested in adherence assays. Each strain tested showed the characteristic adherence pattern corresponding to the pathotype identified (Figure 1). The strains classified as aEPEC or ETEC showed an AA (aggregative adherence) adherence pattern, while hybrid ones showed both associated adherence patterns: AA and diffuse. This can be caused by the presence of adhesins related to the pathotype that might be acquired during its evolutionary history through horizontal gene transfer mechanisms.

### 2.6. Virulence Genes Are Differentially Spread between E. coli Strains

An extended scheme based on presence- for virotyping was performed for each strain. Eleven strains were classified into the D virotype, one was classified into G virotype, and three strains did not harbor the investigated genes, grouping them into virotype 0. All these virotypes have been previously reported [22]. Interestingly, the remaining 75 strains carried enough differential gene arrangements to classify them into three new virotypes: H, which can present *iroN*, *sat*, *papGII*, *cnfI*, *papG*II, *cdtB*; the K1 variant of the capsule *neuC*, virotype I, which presented *sat*, *ibeA*, *hlyA*, *cdtB*; and the K1 variant of the capsule *neuC*, virotype J, which harbors *sat, papG*III, and *cdtB* (Table 2).

### 2.7. The E. coli Strains Isolated from Healthy Donors Showed Important Biofilm Production Traits

Biofilm formation ability is considered an essential trait for pathogenic bacteria. Here, we tested this characteristic for each strain. The results obtained showed a high number of biofilm producer strains (Figure 2). Twenty strains produced at least one order of magnitude more biofilm than control strains; interestingly, four of the strains (Ec-17.4, Ec-27.4, Ec-35.1, and Ec-44.5) showed the highest rate of biofilm production. It could be interesting to investigate what determinants are present in those strains that might be related to this phenotype. 

## 3. Discussion

*E. coli* is one of the predominant species in the intestinal microbiota of humans and animals. Commonly, intestinal strains of *E. coli* are not pathogenic and coexist in harmony with the host, with some even benefiting the host by synthesizing cofactors and protecting it from invasion by pathogenic microorganisms. However, some strains are pathogenic and can cause enteric infections (diarrhea, dysentery, hemorrhagic colitis, hemolytic uremic syndrome) or extraintestinal infections (urinary tract infections, bacteremia or septicemia, meningitis, peritonitis, abscesses, mastitis, and lung and wound infections) [24]; those are known as pathotypes. The most prevalent pathotype worldwide is EPEC, followed by ETEC. Here, we report a high prevalence of atypical EPEC strains (since they do not present the *eae* gene) and the presence of three hybrid pathotypes. In this project, diarrhea-producing *Escherichia coli* strains were found from apparently healthy carriers from H. Caborca, Sonora, Mexico, and the diarrheagenic pathotype of atypical enteropathogenic *E. coli* was found to be more prevalent in the isolates obtained. Three new hybrid pathotypes were also found, which were aEPEC/ETEC, aEPEC/DAEC, and aEPEC/ETEC/DAEC. However, this finding is of utmost importance, since the isolates obtained in this study belonged to people who appeared healthy, but showed the presence of pathotypes associated with a diarrheal condition. To our knowledge, this is the first report of hybrid strains sharing genetic determinants of three diarrheagenic pathotypes. 

Another goal of this study was to determine the co-occurrence of virulence and resistance; the strains analyzed here harbor several genetic arrangements, with fifteen belonging to previously reported virotypes [23]. Nevertheless, a high number of strains showed a virulence genes distribution that does not accord with those previously proposed. The new proposed virotypes presented mainly iron capture systems, but also toxins or invasins, which can provide the ability to cause damage to the host cell. In addition, strains that carry such genes can act as gene reservoirs for other bacteria, even those of different species. Another important trait associated with virulence is the ability of the strains to develop biofilm. Twenty strains were highly biofilm-producing. Interestingly, three of these strains showed a gene distribution that classifies them as virotype H strains, which carry seven virulence genes. Of the genes in this virotype, *cnfI* and *cdtB* stand out, as they are related to several types of carcinoma [22] and promote cell cycle arrest [25], respectively. This gene burden may represent an important health risk.

Seven strains carrying the *bla*_CTX-M1_ gene were identified, two of which belong to virotype H, one to virotype I, and nine to virotype J. 

CTX-M class β-lactamases encoded by *bla*_CTX-M_ genes are the most prevalent type in bacteria in clinical isolates and have become an area of concern due to their high spread [26,27]. These genes are often encoded within conjugative plasmids that may also harbor virulence genes, constituting an important risk of horizontal gene transfer to bacteria resident in different microenvironments [28,29,30]. The contribution of the CTX-resistance associated genes was also evaluated; interestingly four strains from the same donor, showed the same resistance level for the three cephalosporines tested, even they carry one or more genes; on the other hand, the remaining three strains were more heterogeneous since they showed differential behavior for MIC. This could be explained by the presence of other resistance determinants that were not investigated in this study and which could be involved in the resistance traits of the strains.

Seven of the strains analyzed in this study carry the *bla*_CTX-M1_ gene and additionally present virulence genes; potentially, these may represent a risk for the patient as soon as conditions favor it, or they may act as reservoirs of mobilizable genetic elements that may eventually be acquired by bacteria in susceptible patients, generating host damage and representing a therapeutic challenge in the clinical course of the disease. The presence of these strains also implies an important epidemiological risk because of their potentially easy distribution, and might be play an important role in community-acquired infections.

We also observed that the strains analyzed presented high biofilm production capacity, as they are at the production level of *S. aureus* ATCC 25923, which is considered a strong biofilm producer [31]. Interestingly, four of these strains produced higher levels than the control strain and could represent an important feature in the pathogenic process as well as the role that biofilm represents in terms of antimicrobial resistance. An interesting finding was the identification of four of the strains as hyper-producers of biofilm; this may be associated with high levels of antibiotic resistance in an infectious process, as has been previously reported [32,33,34,35,36,37,38,39,40,41,42,43,44,45,46,47,48,49,50]. However, further testing will be necessary to prove such an assertion.

## 4. Conclusions

Strains of *Escherichia coli* harboring diarrheagenic genetic determinants represent an important health risk, either by causing damage to the host or by acting as reservoirs of virulence genes. Additionally, the co-occurrence of genes coding for resistance mechanisms implies an important therapeutic challenge. Asymptomatic hosts can carry these microorganisms without manifesting any signs or symptoms; however, they can act as transmitters to susceptible hosts, complicating treatment options.

Finally, the finding of multidrug-resistant strains with pathogenic potential in healthy donors is important because of the possible dissemination of these strains or the genes they harbor, making it necessary to reinforce domestic hygienic habits.

## 5. Materials and Methods

### 5.1. Sampling

Samples of stool from 50 healthy donors were collected in sterile plastic containers. Each donor was interviewed to ensure they were not under antibiotic treatment at least five days before the sample collection, nor presenting with diarrhea or gastric distention. Informed consent was signed. Samples and personal information were handled following the Declaration of Helsinki, Ethical Principles for Medical Research Involving Human Subjects 2013.

### 5.2. Isolation and Biochemical Characterization of E. coli

After visual inspection looking for blood, mucus, or pus, the samples were inoculated on agar MacConkey plates (BD Bioxon^®^, Beckton Dickinson de, Mexico City, Mexico) and incubated for 18 h at 37 °C. After that, at least five lactose-positive and three lactose-negative colonies were recovered for biochemical identification and freeze storage. Biochemical identification was performed using IMViC tests and assimilation of urea, lysine decarboxylation-deamination, and sugar fermentation (glucose, lactose, and sucrose)

### 5.3. Genomic DNA Extraction

Genomic DNA was obtained using the boiling method. A 24 h culture on Luria Bertani (LB) agar plates of each sample was harvested in 500 µL of nuclease-free water, boiled for 15 min, and centrifuged for 3 min at 13,000 rpm. The supernatant was recovered into a new tube. DNA concentration was adjusted to 200 ng/mL. 

### 5.4. Molecular Identification of E. coli

The strains positively identified as *E. coli* and those identified as *Leclercia adecarboxylata* were tested with conventional PCR using GoTaq Green MasterMix (Promega Corporation, Madison, WI, USA) and specific primers for the *ybbW* gene (Appendix A). The amplicon was resolved on a 1% agarose gel electrophoresis.

### 5.5. Phylogenetic Group Determination

The phylogenetic group of the identified strains was determined using the scheme previously proposed by Clermont et al. (Appendix A Appendix A) [19].

### 5.6. Antibiotic Agar Dilution and ESBL Genes Detection

The resistance profile of each sample was determined by the agar dilution method according to the CLSI (CLSI, 2021). The antibiotic concentration chosen was at least the intermediate level suggested by the M100 manual of CLSI. Two microliters of a preculture from each strain were deposited (drop-plated) on agar MacConkey plates supplemented with gentamycin (8 µg/mL), amikacin (32 µg/mL), sulfamethoxazole (4 µg/mL), nalidixic acid (16 µg/mL), ciprofloxacin (0.5 µg/mL), ampicillin (16 µg/mL), ceftriaxone (2 µg/mL), cefoxitin (16 µg/mL), cefotaxime (2 µg/mL), meropenem (2 µg/mL), nitrofurantoin (64 µg/mL), fosfomycin (128 µg/ml), chloramphenicol (16 µg/ml), or tetracycline (8 µg/ml) individually [24]. The plates were incubated for 18 h at 37 °C; growth was evaluated as resistant, while no growth was considered as sensitive.

Detection of *bla*_CTX-M_ genes was performed with conventional PCR using GoTaq Green MasterMix (Promega Corporation, Madison, WI, USA), and specific primers (Appendix A). The amplicons were resolved on a 1% agarose gel electrophoresis.

### 5.7. Determination of Minimum Inhibitory Concentration (MIC)

Each strain that showed the presence of resistance genes was tested for MIC according to CLSI specifications [21]. Dilutions of ceftriaxone (CRO), cefotaxime (CTX), and cefoxitin (FOX) in several concentrations were used (256 µg/mL, 128 µg/mL, 64 µg/mL, 32 µg/mL, 16 µg/mL, 8 µg/mL and 4 µg/mL) in 100 µL of Müeller Hinton broth distributed in a 96-well plate. A 1:20 dilution of a 0.5 McFarland standard of an adjusted preculture of each strain was added to the well. After 18 h of incubation at 37 °C, the plate was read for inhibition of growth. Each strain was triple tested.

### 5.8. Determination of Diarrheagenic Pathotypes

Conventional PCRs were performed searching for genetic pathotype markers: *bfpA* gene for EPEC identification, *eae* gene for EPEC/EHEC identification, *stx1* and *stx2* for EHEC/STEC identification, thermolabile and thermostable toxins for ETEC identification, *daaE* and *afa/draBC* genes for DAEC identification, pCVD432 for EAEC identification, and *iaI* for EIEC identification (Appendix A).

### 5.9. Adherence Assay

One strain of each pathotype detected was tested for adherence patterns. HeLa cells were seeded on tissue culture plates in Dulbecco’s Modified Eagle Media (DMEM) (Gibco, Waltham, MA, USA) supplemented with 10% fetal bovine serum (FBS) (Gibco, Waltham, MA, USA) at 37 °C in 5% CO_2_ until sub-confluence. Then, 5 mL of FC Wash solution with 0.25% trypsin solution was added, incubated for 3 min at 37 °C, and decanted. Fresh DMEM + 10%FBS was added. Cells were adjusted to 5 × 10^4^/mL and 425 µL were seeded on each well of an eight-well Millicel^®^EZ slides (Merck Millipore Ltd., County Cork, Ireland). The slide was then incubated overnight at 37 °C in 5% CO_2_. HeLa cell monolayers were washed with sterile PBS. After washing, 88.5 µL of bacterial suspension (50,000,000 UFC/mL) in DMEM were added to each well (MOI 1:20, HeLa: Bacteria) and incubated for two h at 37 °C in 5% CO_2_. After incubation, wells were washed twice with PBS. Methanol was used to fix cell monolayers for 10 min and samples were stained with Giemsa. The adhesion patterns were observed [35].

### 5.10. Virotyping

To determine virulence gene loads of each sample, the virotyping scheme proposed by Nicolas-Chanoine et al. for ST131 classification and extended by Barrios-Villa et al. was implemented. Eleven virulence factor (VF) genes were determined for the *E. coli* isolates using a scheme that identifies *pap*, *cnfI*, *sat*, *iroN*, *cdtB*, *afa/draBC*, *ibeA*, *hlyA*, and *kpsM*II genes distributed into seven established virotypes of *E. coli* ST131 (A–G) [22,23]. Specific primers and conditions are listed in Table 1. Amplicons were resolved on a 1% agarose gel electrophoresis.

### 5.11. Biofilm Formation Assays

The ability to form biofilms was determined in a 96 wells plate. Bacteria were incubated in LB broth for 24 h at 37 °C and biofilm formation was determined according to protocols previously reported [36]. As positive controls, *E. coli* ATCC 25922 and *Staphylococcus aureus* ATCC 25923 were implemented.

## Figures and Tables

**Figure 1 antibiotics-11-00833-f001:**
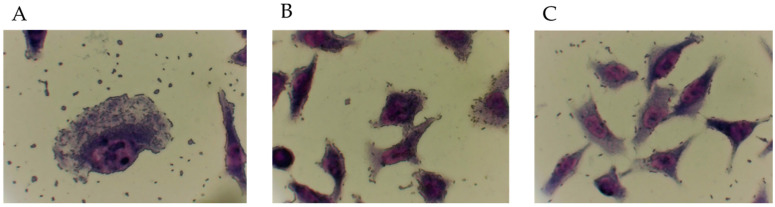
Adherence patterns identified in this study. (**A**), localized adherence (present in strains Ec-9.3, Ec-37.1, Ec-41.3, Ec-44.2, Ec-36.1, Ec-36.3, Ec-31.3, and Ec-37.4); (**B**), aggregative adherence (observed in strains Ec-44.5, Ec-45.3, Ec-46.2, Ec-50.1, and Ec-50.2); (**C**), diffuse adherence (present in strains Ec-36.1 and Ec-36.3).

**Figure 2 antibiotics-11-00833-f002:**
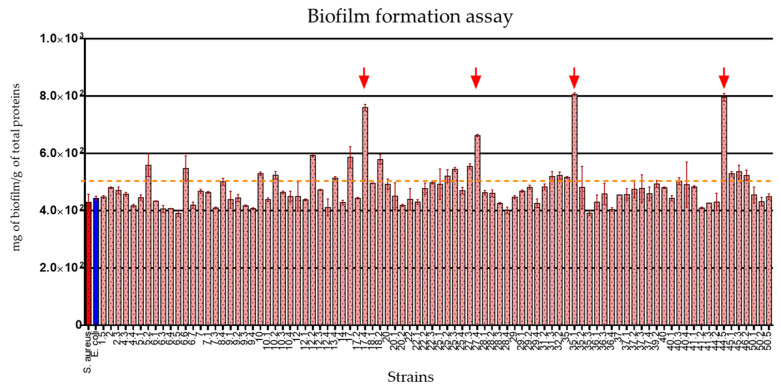
Biofilm formation assay. The dotted line represents 50% of the measurement parameter (1 g of total protein). The average of three assays was interpolated to a bovine serum albumin calibration curve. Red arrows indicate the four strains that showed a higher biofilm production, higher than 800 mg/g of total protein. *S. aureus* ATCC25923 (red bar) and *E. coli* ATCC25922 (blue bar) were used as positive controls.

**Table 1 antibiotics-11-00833-t001:** Characteristics of strains isolated in this study.

Strain	Phylogroup	Pathotype	Resistotype	ESBL Genes	Minimum Inhibitory Concentration (µg/mL)
*Ec-1.5*	A	NA	TE, DCI, FOX, AMP	---	---
Ec-2	B2	NA	TE, DCI, AMP	---	---
Ec-2.3	B2	NA	TE, DCI, AMP	---	---
**Ec-4.3**	B2	NA	TE, DCI, AMP	---	---
**Ec-4.4**	B2	NA	DCI, AMP	---	---
**Ec-5.1**	C	NA	DCI, AMP	---	---
**Ec-5.2**	A	NA	DCI, AMP	---	---
**Ec-6.1**	C	NA	DCI, FOX, CRO, AMP	---	---
**Ec-6.3**	C	NA	DCI, AMP	---	---
Ec-6.4	Unknown	NA	DCI, AMP	---	---
Ec-6.5	B2	NA	TE, C, DCI, FOX, CRO, AMP	---	---
Ec-6.6	Unknown	NA	TE, C, DCI, AMP	---	---
**Ec-6.7**	B2	NA	TE, DCI, AMP	---	---
Ec-7	B2	NA	TE, DCI, AMP	---	---
Ec-7.1	D	NA	DCI, AMP	---	---
Ec-7.3	B2	NA	FOS, DCI, AMP	---	---
Ec-8.4	B1	aEPEC	DCI, AMP	---	---
Ec-9.1	A	NA	FOS, DCI, FOX, CRO, AMP	CTX-M1, CTX-M2	256 (FOX, CTX, CRO)
Ec-9.2	B2	NA	FOS, DCI, FOX, CRO, AMP	CTX-M1, CTX-M3	256 (FOX, CTX, CRO)
Ec-9.3	A	aEPEC	DCI, FOX, CRO, AMP	CTX-M1	256 (FOX, CTX, CRO)
Ec-9.4	A	NA	DCI, FOX, CRO, AMP	CTX-M1	256 (FOX, CTX, CRO)
Ec-10	C	aEPEC	DCI, AMP	---	---
Ec-10.1	B2	aEPEC	FOS, DCI, AMP	---	---
Ec-10.2	A	aEPEC	DCI, AMP	---	---
Ec-10.3	B2	NA	DCI, AMP	---	---
Ec-10.4	B2	aEPEC	DCI, AMP	---	---
Ec-12	A	NA	DCI, AMP	CTX-M1	128 (FOX, CTX, CRO)
Ec-12.1	B2	NA	DCI, FOX, CRO, AMP	CTX-M1	128 (FOX, CTX), 256 (CRO)
Ec-12.2	B2	NA	DCI, FOX, CRO, AMP	CTX-M1	64 (FOX, CTX), 256 (CRO)
Ec-12.3	B2	NA	DCI, FOX, CRO, AMP	---	---
Ec-12.4	A	NA	DCI, CRO, AMP	---	---
Ec-13.4	C	NA	DCI, FOX, AMP	---	---
Ec-14	F	aEPEC	DCI, AMP	---	---
Ec-17	B2	aEPEC	SENSITIVE	---	---
Ec-17.2	B2	ETEC	TE, DCI, AMP	---	---
Ec-17.4	B2	aEPEC	TE, DCI, AMP	---	---
**Ec-18.1**	C	NA	TE, DCI, AMP	---	---
**Ec-18.2**	B2	aEPEC	TE, DCI, AMP	---	---
Ec-20	B2	aEPEC	TE, DCI, AMP	---	---
Ec-20.1	B2	aEPEC	TE, DCI, AMP	---	---
**Ec-20.2**	B2	aEPEC	TE, DCI, AMP	---	---
**Ec-22**	F	NA	DCI, AMP	---	---
**Ec-22.1**	B2	NA	DCI, AMP	---	---
**Ec-22.2**	B2	NA	DCI	---	---
**Ec-22.3**	B2	aEPEC	TE, DCI, AMP	---	---
**Ec-25.1**	F	NA	DCI, AMP	---	---
**Ec-25.2**	A	aEPEC/ETEC	DCI, AMP	---	---
*Ec-25.3*	B2	aEPEC	DCI, AMP	---	---
**Ec-25.4**	A	aEPEC	DCI, AMP	---	---
Ec-27.3	A	NA	TE, DCI, AMP	---	---
Ec-27.4	A	aEPEC	TE, FOS, DCI, AMP	---	---
**Ec-28.1**	B2	NA	TE, DCI, AMP	---	---
Ec-28.2	Unknown	aEPEC	TE, DCI, AMP	---	---
Ec-28.3	Unknown	aEPEC	TE, DCI, AMP	---	---
**Ec-28.4**	A	aEPEC	TE, DCI, AMP	---	---
**Ec-29**	A	aEPEC	TE, DCI, AMP	---	---
Ec-29.1	A	aEPEC	TE, DCI, AMP	---	---
**Ec-29.2**	C	aEPEC	TE, DCI, FOX, AMP	---	---
Ec-29.4	C	aEPEC	TE, DCI, DOX, AMP	---	---
Ec-31.2	B2	aEPEC	TE, DCI, AMP	---	---
Ec-31.3	A	ETEC	TE, DCI, AMP	---	---
Ec-32.2	B2	NA	TE, DCI, AMP	---	---
Ec-35	B2	aEPEC	TE, FOS, DCI, AMP	---	---
Ec-35.1	Unknown	aEPEC	TE, C, FOS, DCI, AMP	---	---
Ec-35.2	C	aEPEC	TE, C, FOS, DCI, AMP	---	---
Ec-35.3	C	aEPEC	TE, DCI, AMP	---	---
Ec-36.1	B2	aEPEC/DAEC	FOS, DCI, AMP	---	---
Ec-6.3	B2	aEPEC/ETEC/DAEC	DCI	---	---
Ec-36.4	B2	aEPEC/ETEC	DCI	---	---
Ec-37	B2	aEPEC	TE, C, DCI, AMP	---	---
Ec-37.1	F	aEPEC	TE, C, DCI, AMP	---	---
**Ec-37.2**	B2	aEPEC	DCI, AMP	---	---
**Ec-37.3**	B2	NA	TE, C, DCI, AMP	---	---
Ec-37.4	F	ETEC	TE, C, DCI, AMP	---	---
Ec-39.2	D	NA	C, DCI, AMP	---	---
Ec-40	B2	NA	DCI, AMP	---	---
Ec-40.1	C	aEPEC	TE, DCI, AMP	---	---
Ec-40.3	F	aEPEC	TE, DCI, AMP	---	---
Ec-40.4	B2	aEPEC	TE, DCI, AMP	---	---
**Ec-41.1**	B1	NA	DCI, AMP	---	---
Ec-41.2	B1	NA	C, DCI, AMP	---	---
Ec-41.3	B1	aEPEC	DCI, AMP	---	---
Ec-44.2	B1	aEPEC	TE, C, DCI, AMP	---	---
Ec-44.5	B1	NA	TE, DCI, AMP	---	---
Ec-45.1	B1	NA	DCI, AMP	---	---
Ec-45.3	B1	NA	DCI, AMP	---	---
Ec-46.2	F	NA	DCI, AMP	---	---
Ec-50.1	B2	NA	TE, DCI, AMP	---	---
Ec-50.2	B2	NA	DCI, FOX, CRO, AMP	---	---
Ec-50.5	B2	NA	TE, DCI, AMP	---	---

Strains named with the same integer number correspond to isolates from the same patient. Underlined numbers correspond to strains classified into virotype D, bold numbers correspond to strains classified into virotype H, italicized numbers correspond to strains classified into virotype I, shaded numbers correspond to strains classified into the “0” virotype, and the remaining strains were classified into virotype J. NA, not associated, negative PCR for Diarreaghenic *E. coli* genetic marker; (---), absence of *bla* genes. TE, tetracycline; DCI, nitrofurantoin; FOX, cefoxitin; AMP, ampicillin; CRO, ceftriaxone; C, chloramphenicol; FOS, fosfomycin.

**Table 2 antibiotics-11-00833-t002:** Virulence coding genes present in *E. coli* strains isolated from healthy donors from H. Caborca, Mexico.

Virotype	*afa/draBC*	*afa* Operon	*iroN*	*satA*	*ibeA*	*papG*II	*cnfI*	*hlyA*	*papG*III	*cdtB*	*neuC-*KI
A ^a^	+	+	−	±	−	−	−	−	−	−	−
B ^a^	−	−	+	±	−	−	−	−	−	−	−
C ^a^	−	−	−	+	−	−	−	−	−	−	−
D ^a^	±	±	±	±	+	−	±	±	±	±	±
E ^a^	−	−	−	+	−	+	+	+	−	−	−
F ^a^	−	−	+	−	−	+	+	−	−	−	−
G ^a^	−	−	−	+	−	+	−	−	−	−	−
H *	−	−	±	±	−	+	±	−	±	±	±
I *	−	−	−	+	+	−	−	±	−	+	+
J *	−	−	−	+	−	−	−	−	+	+	−
0	−	−	−	−	−	−	−	−	−	−	−

Notes: ^a^, Major virotypes [23]; an asterisk (*) indicates new proposed virotypes. +, positive PCR; −, negative PCR; ±, variable PCR. *afa/draBC*, afa/Dr adhesin; *afa* operon, FM955459; *iroN*, catecholate iron receptor; *sat*, autotransporter secreted toxin; *ibeA*, epithelial brain invasion protein; *papG*II, papG allele II; *cnf1*, cytotoxic necrotizing factor type 1; *hlyA*, α-hemolysin; *papG*III, papG allele III; *cdtB*, cytolethal distention toxin; *neuC*-K1, group II capsule K1 variant.

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
