# Peer review of "Characterization of Diarreaghenic Escherichia coli Strains Isolated from Healthy Donors, including a Triple Hybrid Strain"

_antibiotics, 2022, doi:10.3390/antibiotics11070833_

Round 1
Reviewer 1 Report
The work of Mendez-Moreno et al is interesting.
The title is too long. I will suggest the authors to change it.
Please highlight the clinical significance of the work in a separate paragraph of discussion section.
Author Response
Dear Reviewer, I would like to tank you for your observations which are really valuable to improve the manuscript. I am indicating the changes made according with your suggestions.
The title is too long. I will suggest the authors to change it. The change was made.
Please highlight the clinical significance of the work in a separate paragraph of discussion section. The suggestion was addressed and it is located between lines 290-297.
Reviewer 2 Report
Overall, the paper is well written, and the subject content interesting. Showing that E. coli strains with various pathotypes and of varying antibiotic resistant being harbored in healthy donors is of significant importance.
The methodology used in this work is appropriate. I only have a few suggestions that should be addressed before publication. The last sentence on lines 29 and 30 needs a small edit. I think the sentence needs to end with something like the following: ....was superior to that of the E. coli strains used as positive controls. The in all strains part seems not to be needed.
For the most part, the methods and the data are strong and support the issues brought up in the discussion section. All of the methods used were appropriate for the study. The only issue I would like to see addressed in the results section is the antibiotic resistance testing and reporting. The testing done to report the resistotypes listed in table 1 is fine, however, I think more could be done to determine the level of resistance to the particular antibiotics and some statistics to show that the level of resistance to the antibiotics are significant.
It is important to know the level of resistance these particular isolates display. Just because they carry antibiotic resistance genes and they grow on a standard level of an antibiotic only scratches the surface of this important finding. Multiple factors can influence the level of antibiotic resistance and the expression of antibiotic resistance genes. A little more investigation into the degree of the antibiotic resistance in the E. coli isolates is important and is something I would recommend prior to publication.
Author Response
Dear Reviewer, I would like to thank you for your observations which are really valuable to improve the manuscript. I am indicating the changes made according with your suggestions.
The last sentence on lines 29 and 30 needs a small edit. I think the sentence needs to end with something like the following: ....was superior to that of the E. coli strains used as positive controls. The in all strains part seems not to be needed. The suggestion was addressed and the change was made as follows "was superior to that of the E. coli and Staphilococcus aureus strains used as positive controls".
The only issue I would like to see addressed in the results section is the antibiotic resistance testing and reporting. Your suggestion was attended, we performed Minimum Inhibitory Concentration test and the table 1 was improved, the results are also included between lines 109-15 and in the discussion section between lines 283-289.
Thank you in advance.
The authors